# PCA-Driven Multivariate Trait Integration in Alfalfa Breeding: A Selection Model for High-Yield and Stable Progenies

**DOI:** 10.3390/plants14182906

**Published:** 2025-09-18

**Authors:** Zhengfeng Cao, Jiaqing Li, Huanwei Lei, Mengyu Yan, Qianxi Wang, Runqin Ji, Siqi Zhang, Xueyang Min, Zhengguo Sun, Zhenwu Wei

**Affiliations:** 1College of Animal Science and Technology, Yangzhou University, Yangzhou 225009, China; caozhengfeng@yzu.edu.cn (Z.C.);; 2Institute of Grassland Science, Yangzhou University, Yangzhou 225009, China; 3College of Agro-Grassland Science, Nanjing Agricultural University, Nanjing 210095, China; sunzg@njau.edu.cn

**Keywords:** *Medicago sativa* L., alfalfa breeding, principal component analysis, multivariate selection, trait trade-offs, genetic gain

## Abstract

Breeding improvement in alfalfa (*Medicago sativa* L.) is often constrained by the complexity of agronomic traits and trade-offs among yield-related characteristics. Conventional single-trait selection rarely captures the full range of phenotypic variation or the interactions among traits. To address this, we developed a principal component analysis (PCA)-based framework for multivariate selection in hybrid breeding. Six yield-related traits—plant height, branch number, fresh/hay yield ratio (FHR), leaf/stem ratio (LSR), multifoliolate leaf frequency, and dry weight per plant—were quantified in two parental lines and their F_1_/F_2_ generations. PCA identified three principal components (PC1–PC3) with eigenvalues >1, explaining 71.14% of the total phenotypic variance: PC1 (32.43% variance) was predominantly loaded with positive contributions from dry weight per single plant, height, and branches, biologically representing overall plant vigor and biomass accumulation; PC2 (21.77% variance) showed strong negative loadings for LSR, capturing architectural trade-offs between stem dominance and leaf production; PC3 (16.94% variance) had positive loadings on multifoliolate leaf rate and fresh/dry ratio, embodying quality and physiological resilience traits. Based on PCA scores, a composite selection index was constructed, and the top 31.1% of F1 hybrids were selected. Their F_2_ progeny showed significant improvements in dry weight (+15.56%, *p* < 0.01), multifoliolate leaf frequency (+74.78%, *p* < 0.001), and reduced FHR (–8.2%, *p* < 0.05), accompanied by lower yield decline (−7.2% versus −14.1% in controls). These results show that PCA-based multivariate selection effectively balances trait trade-offs, enhances intergenerational stability, and improves selection efficiency. This framework offers a practical tool for alfalfa breeding.

## 1. Introduction

Principal component analysis (PCA), a widely used dimensionality reduction technique, is extensively applied in crop breeding for integrative analysis of multiple traits within phenomics. It effectively handles high-throughput phenotypic data such as near-infrared spectroscopy (NIRS) [1] or hyperspectral imaging [2], accelerating breeding cycles by extracting key patterns from complex datasets. PCA converts high-dimensional phenotypic data (such as spectral absorption values) into a small number of principal components (PCs), capturing the major sources of variation and thereby reducing computational burden; numerous studies have further demonstrated its effectiveness in consolidating yield-related traits and in identifying genotypes associated with desirable comprehensive agronomic profiles [3,4]. For instance, PCA has been utilized to integrate key agronomic traits—such as plant height, panicle number, and yield per plant—across a wide range of crops. This approach has been successfully applied in the breeding of rice (*Oryza sativa* L.) [5], fingerroot (*Boesenbergia rotunda* L.) [6], sweet potato (*Ipomoea batatas* [L.] Lam) [7], sorghum (*Sorghum bicolor* [L.] Moench) [8], cotton (*Gossypium hirsutum* L.) [9], and maize (*Zea mays* L.) [10], leading to improved selection efficiency and breeding outcomes.

PCA is particularly powerful in revealing synergistic effects and trade-offs among multiple traits. Some crops exhibit positive correlations, such as plant height and dry weight [11,12]. In contrast, others display negative associations, for example between cyanidin content in alfalfa and neutral detergent fiber content [13]. Through PCA, these interactions can be quantitatively characterized, providing breeders with insights into complex trait networks and helping to minimize conflicts among traits during selection. Beyond trait integration, PCA has also been shown to enhance predictive capacity in breeding, thereby shortening breeding cycles. By evaluating multiple traits simultaneously, PCA effectively elucidates genetic structure and reproductive barriers among populations with varying ploidy levels, thereby revealing patterns of isolation and differentiation [14]. Moreover, by removing redundant information and reducing variable dimensionality [15], PCA facilitates the rapid identification of promising interspecific hybrids from phenomic markers, reducing the need for extensive field trials and accelerating breeding cycles by prioritizing individuals with desirable trait combinations [16].

Despite its advantages, PCA-based breeding approaches still face certain limitations and challenges. One key issue is balancing the number of traits with an adequate sample size. Although PCA reduces dimensionality, its performance depends on sufficient input data. Inadequate sample sizes may prevent the model from capturing critical trait variation, thus undermining reliability [17]. To address this, some researchers have proposed integrating genomic data to increase the effective sample size and improve predictive accuracy. Moreover, PCA is inherently linear and may not capture nonlinear relationships present in biological data [18].

Another challenge lies in the interpretability and potential bias of principal components. While PCA summarizes data through the extraction of components, these components are often abstract linear combinations of the original variables and may lack clear biological meaning [19,20]. This makes it difficult to directly relate them to breeding targets such as yield or disease resistance. Improving the interpretability of principal components and aligning them with practical breeding objectives is a pressing research need. Additionally, selection bias may occur if important but seemingly insignificant traits are discarded during dimensionality reduction, potentially leading to overfitting, especially in models relying on fixed trait weights.

Environmental effects also play a crucial role in trait expression. Variability in environmental conditions can significantly influence phenotypic traits and reduce the stability and reliability of PCA models. As such, incorporating environmental data into PCA can help identify which environmental factors are most closely associated with genetic variation and allow breeders to track how such variation responds across different environmental contexts [21].

In summary, PCA is a powerful and widely adopted tool in plant breeding, particularly effective for multi-trait integration and breeding cycle reduction. Nonetheless, current applications still face challenges such as optimizing trait-to-sample ratios, improving interpretability, and accounting for environmental influences. Future research that combines PCA with genomic data, high-dimensional phenotyping, and environmental adaptability analysis may enhance its accuracy and robustness, making it an even more effective tool for crop improvement. In this study, we propose a PCA-driven selection model aimed at optimizing hybrid breeding in alfalfa (*Medicago sativa* L.). Traditional breeding approaches often focus on individual traits—such as plant height or dry weight—while overlooking interactions among agronomic characteristics. The goal of our research is to develop an integrated framework based on PCA to capture inter-trait relationships, improve biomass yield and stability, and ultimately enhance selection efficiency in alfalfa breeding.

## 2. Results

### 2.1. Agronomic Trait Characterization in Parental Lines and F1 Hybrids

Six yield-related traits—absolute plant height, branch number, FHR, LSR, multifoliolate leaf frequency, and dry weight per plant—were quantified at the initial flowering stage in parental lines (PL34HQ, Huaiyin) and F1 hybrids (*n* = 90) (Appendix A). Significant differences (*p* < 0.01) were observed between parents for five traits (plant height, branch number, FHR, multifoliolate trait frequency and dry weight), while LSR showed no significant divergence (*p* > 0.05). F_1_ hybrids exhibited intermediate trait values between parental extremes (Table 1.).

### 2.2. Correlations Among Agronomic Traits in the F_1_ Generation of Alfalfa Crosses

Trait correlations in F_1_ hybrids revealed strong positive associations between dry weight and plant height (ρ = 0.35, *p* < 0.001), branch number (ρ = 0.30, *p* < 0.01), and LSR (ρ = 0.45, *p* < 0.001), alongside a negative correlation with FHR (ρ = −0.26, *p* < 0.05). Plant height and branch number were also highly correlated (ρ = 0.49, *p* < 0.001) (Figure 1).

### 2.3. Principal Component Analysis (PCA) for Dimensionality Reduction

PCA of six agronomic traits extracted three principal components (PC1–PC3) with eigenvalues >1, cumulatively explaining 71.14% variance (PC1: 32.43%, PC2: 21.77%, PC3: 16.94%) (Table 2, Figure 2). Component coefficients were derived by dividing initial factor loadings (Table 3) by the square root of corresponding eigenvalues (Table 2). The scree plot (Appendix A) further confirmed the appropriateness of retaining these three components based on the eigenvalue >1 criterion and the elbow method.

The final three principal components obtained are as follows:*F*_1_ = 0.523*X*_1_ + 0.463*X*_2_ − 0.295*X*_3_ + 0.294*X*_4_ − 0.176*X*_5_ + 0.556*X*_6_(1)*F*_2_ = 0.216*X*_1_ + 0.543*X*_2_ − 0.049*X*_3_ − 0.704*X*_4_ + 0.346*X*_5_ − 0.200*X*_6_(2)*F*_3_ = −0.395*X*_1_ − 0.070*X*_2_ − 0.450*X*_3_ + 0.101*X*_4_ + 0.705*X*_5_ + 0.360*X*_6_(3)
where *X*_1_–*X*_6_ represent standardized values of plant height, branch number, FHR, LSR, multifoliolate trait frequency, and dry weight, respectively.

The loading plot illustrates the contribution of six agronomic traits to the first three principal components (PC1–PC3), which collectively explain 71.14% of the total variance.

Trait weights were calculated by Trait weights were obtained by multiplying the loading coefficients of each trait by the variance contribution of the corresponding principal component, and then summing the weighted values across *F*_1_–*F*_3_ (Formular (1)–(3)). Finally, the result was normalized by the total variance explained by these three components (Table 2). For example, the weighting factor for plant height was calculated as(4)0.523×32.435+0.216×21.771−0.395×19.93771.143=0.210
which was then incorporated into the final selection index:*Y* = 0.210Z*X*_1_ + 0.360Z*X*_2_ − 0.257Z*X*_3_ − 0.057Z*X*_4_ + 0.194Z*X*_5_ + 0.278Z*X*_6_(5)

(Z*X*_1_, Z*X*_2_, Z*X*_3_, Z*X*_4_, Z*X*_5_, Z*X*_6_: Represent the standardized plant height, number of branches, FHR, stem-to-foliage ratio, leafy ratio and dry weight values, respectively).

### 2.4. Selection of Elite Hybrids Using a Multivariate Approach

The model ranked 90 F_1_ hybrids by composite scores (Table 4), selecting the top 28 genotypes (combined scores >1) for subsequent crosses.

### 2.5. Validation of Selection Efficacy in F_2_ Progenies

Compared to unselected populations, F_2_ progeny from elite F_1_ hybrids exhibited a 15.56% higher dry weight (*p* < 0.01), a 74.78% increase in the frequency of the multifoliolate trait (*p* < 0.001), an 8.2% reduction in FHR (*p* < 0.05), and an attenuated yield decline (−7.2% vs. −14.1% in controls; Table 5).

## 3. Discussion

This study demonstrates that a principal component analysis (PCA)-driven selection framework can effectively integrate multiple agronomic traits in alfalfa breeding and translate this integration into tangible genetic gains. Traditional single-trait selection often underestimates the complexity of yield-related traits, which are shaped by multiple, interacting morphological and physiological factors [22,23,24]. Specifically, as illustrated in the 3D PCA biplot (Figure 2), PC1 (explaining 32.43% of the variance) was predominantly loaded with positive contributions from dry weight per single plant, height, and branches, biologically representing overall plant vigor and biomass accumulation. This component likely reflects integrated physiological processes such as enhanced carbon assimilation and resource allocation toward vegetative growth, which are critical in alfalfa for maximizing forage yield under varying environmental conditions [25,26]. In contrast, PC2 (21.77% variance) showed strong negative loadings for leaf/stem ratio, suggesting it captures architectural trade-offs between stem dominance and leaf production—potentially linked to light interception efficiency and mechanical stability, enabling breeders to select for balanced plant morphology that reduces lodging risks [27,28]. PC3 (16.94% variance), with positive loadings on multifoliolate leaf rate on the main stem and fresh/dry ratio, appears to embody quality and physiological resilience traits, such as leaf complexity for improved digestibility and water retention for stress tolerance. For breeders, this is meaningful as it highlights genotypes with enhanced nutritional value and yield stability, addressing common challenges in forage legumes where multifoliate leaves correlate with higher protein content [29,30]. By selecting based on favorable scores across these PCs, the approach prioritizes holistic trait networks, which explains the superior performance of F_2_ progeny from high-ranking F_1_ hybrids: reinforced beneficial interactions led to immediate agronomic gains and intergenerational consistency. By constructing a composite scoring model based on PCA, we were able to rank and select elite F_1_ hybrids, and subsequently advance them to F_2_ populations. The results provide strong empirical evidence for the power of this multivariate approach: compared with unselected populations, F_2_ progeny derived from elite F_1_ hybrids exhibited 15.56% higher dry weight (*p* < 0.01), a 74.78% increase in multifoliolate trait frequency (*p* < 0.001), a reduced FHR (−8.2%, *p* < 0.05), and attenuated yield decline (−7.2% versus −14.1% in controls). These improvements highlight the ability of PCA-based selection to both enhance agronomic performance and stabilize yield-related traits across breeding cycles.

The F_1_ population provided important insights into the relationships among traits. Correlation analyses revealed that dry weight per plant was positively associated with plant height, branch number, and LSR, but negatively associated with FHR. Such synergistic relationships suggest that biomass accumulation in alfalfa is not determined by any single attribute, but rather emerges from the interaction of multiple traits [31,32]. The PCA model effectively captured these interactions by assigning greater weight to traits contributing positively to yield and down-weighting antagonistic factors [33,34]. This may explain why the F_2_ progeny derived from high-ranking F_1_ individuals outperformed unselected controls: selection favored genotypes in which beneficial trait networks were reinforced, leading to both immediate improvement and potential intergenerational stability.

Interestingly, while most traits differed significantly between parents, LSR remained relatively stable across parental lines, F_1_ hybrids, and F_2_ progeny. Because LSR is a recognized quality indicator in alfalfa [35,36], its stability suggests that this trait may be less sensitive to environmental variation and largely under genetic control [28]. The lack of significant divergence in LSR across generations therefore indicates both heritable consistency and a limited degree of genetic diversity for this trait in the materials studied. This observation underscores the importance of combining stable traits like LSR with more variable, yield-contributing traits when designing composite selection indices [37,38,39].

Our findings also resonate with broader trends in plant breeding. Multi-trait selection strategies, including PCA-based indices and multivariate genome-wide association studies (mvLMM), have been shown to outperform single-trait approaches in crops such as maize and wheat [40]. These strategies not only capture trait correlations more effectively but also enhance the detection of true genetic signals. By applying a similar framework to alfalfa, our study provides one of the first empirical validations that multivariate composite scoring can accelerate breeding progress in forage legumes [41], a crop where yield stability and quality improvement are both essential but often difficult to achieve simultaneously [42]. Research suggests that in forage grass breeding programs, yield traits are often selected first due to their direct impact on economic viability, with quality traits addressed in later stages to balance overall performance [43,44].

Looking forward, the framework established here offers a flexible and scalable approach for future breeding programs. While this study relied on field-based phenotyping of six traits, the integration of high-throughput phenotyping platforms and genomic selection tools [45,46,47] could further enhance the precision and efficiency of multivariate selection. Such integration would allow breeders to simultaneously capture complex trait interactions at both the phenotypic and molecular levels, thereby increasing selection power and reducing breeding cycles. Ultimately, by combining PCA-based indices with emerging technologies, alfalfa breeding can move toward more systematic, data-driven strategies that accelerate genetic gain, stabilize key agronomic traits, and contribute to sustainable forage crop improvement.

## 4. Materials and Methods

### 4.1. Research Design and Materials

This study aimed to establish a principal component analysis PCA-based multi-trait comprehensive evaluation model to screen alfalfa (*M. sativa* L.) hybrid progenies with both high yield and synergistic trait advantages, and to elucidate patterns of intergenerational trait transmission.

The Australian multifoliate alfalfa (*Medicago sativa* L.) line ‘PL34HQ’ [Source: China-Australia Alfalfa Cooperation Project (ASI/1998/026)] was used as the maternal parent, and the local cultivar ‘Huaiyin alfalfa’ (Source: National Animal Husbandry Station, Beijing, China) served as the paternal parent. F_1_ progeny (90 plants) from a cross between PL34HQ and Huaiyin were evaluated, with 28 plants (top 31.1%) selected by the PCA-based index and the remaining 62 classified as unselected. Each group was intermated separately to generate corresponding F_2_ populations (*n* = 90), which were compared to validate the selection model.

The experiment was conducted at the Yang-tzu-chin Experimental Base for Grassland Science, Yangzhou University, using a strip trial design in which every four rows formed an identical strip group, with a row spacing of 15 cm and plant spacing of 10 cm. In early September 2023, seeds were germinated in a controlled growth chamber under conditions of 25 °C with 16 h of light and 22 °C with 8 h of darkness. Once seedlings reached 3–5 cm in height, they were transplanted into trays filled with a 1:1 vermiculite–peat substrate (Pindstrup, Ryomgaard, Denmark). After establishment, seedlings were transplanted row by row onto outdoor ridges (10 cm high). A basal application of compound NPK fertilizer (18-18-18; Stanley San’an, Linyi, China) was made before transplantation. The field trial relied on natural rainfall, with irrigation applied only once immediately after transplantation to promote seedling survival; no further watering was provided. All materials were individually harvested and assessed at the initial flowering stage to ensure uniformity, with each plant in the population measured once to minimize the impact of growth period variation on trait measurements. During the experimental period, January was the coldest month and July the hottest in Yangzhou. In the first year, the average maximum temperature in July was 30.2 °C, and the average minimum temperature was −1.2 °C; in the second year, these values were 35.2 °C and 26.1 °C, respectively. Monthly precipitation ranged from 11.5 to 166.1 mm in the first year and from 35.3 to 625.1 mm from January to July in the second year. Soil properties were as follows: organic matter, 11.89 g kg^−1^; available nitrogen, 88.26 mg kg^−1^; available phosphorus, 6.04 mg kg^−1^; available potassium, 42.33 mg kg^−1^; and pH, 7.34.

### 4.2. Measurement and Analytical Methods for Agronomic Traits

#### 4.2.1. Agronomic Trait Quantification

Measured parameters included:Plant height (PH): the distance between the ground (seedling from cotyledonary node) and the top of the main stem (growing point) after the individual plant has been straightened (cm) [48].Branch number (BN): Total primary branches above root crown (parallel to ground) [48].Multifoliolate trait frequency (MF): The multifoliolate trait frequency (MF) is a species-specific indicator used in alfalfa to evaluate the occurrence of compound leaves with an increased number of leaflets. To determine MF, the total number of compound leaves on a representative branch was recorded, and leaves with four or more leaflets were classified as multifoliolate [48]. The MF was then calculated as the proportion of multifoliolate leaves to the total number of compound leaves on the branch, using the following formula:MF (%) = (Number of compound leaves with ≥ 4 leaflets/Total number of compound leaves) × 100

This trait is commonly used to assess the expression frequency of the multifoliolate phenotype, which is often associated with potential yield improvement and forage quality enhancement.

Fresh weight (FW): Fresh biomass per plant after cutting (g) [49].The leaf/stem ratio (LSR) was determined as the ratio of leaf dry weight to stem dry weight. After harvest, plant samples were manually separated into leaf and stem components. Each component was oven-dried at 65 °C to a constant weight. The LSR was calculated using the following formula:

LSR = Leaf dry weight (g)/Stem dry weight (g)

This ratio serves as an important indicator of forage quality, with higher values generally reflecting improved digestibility and nutritional content [50].

Dry weight (DW): Constant weight after 105 °C enzyme deactivation (30 min) followed by 65 °C drying (g) until a constant weight was achieved [51].The fresh/hay yield ratio (FHR) was calculated as the ratio of fresh biomass weight to dry biomass weight. Fresh weight was measured immediately after harvesting each plant. To determine dry weight, the same plant samples were oven-dried at 65 °C until a constant weight was achieved (typically 48–72 h). The FHR was then calculated using the formula:

FHR = Fresh weight (g)/Dry weight (g)

This ratio reflects the water content of the biomass and serves as an important indicator of forage moisture characteristics and drying efficiency.

#### 4.2.2. Statistical Analysis

**Data processing**: Raw data cleaning and descriptive statistics using spreadsheet software (Excel 2021, Microsoft, Redmond, WA, USA).

**Statistical analyses:** Independent *t*-tests were used to compare differences between two groups (IBM, Armonk, NY, USA). One-way analysis of variance (ANOVA) followed by Duncan’s multiple range test (DMRT) was performed using SPSS 27 (IBM, Armonk, NY, USA) to assess differences among multiple groups (α = 0.05).

**Trait correlations**: Inter-trait associations evaluated using Pearson correlation coefficients (ρ) with α= 0.05.

**Principal component analysis** (**PCA**): Dimension reduction analysis of six standardized traits conducted in Origin 2024 (OriginLab, Northampton, MA, USA), with principal components extracted via Kaiser criterion (eigenvalue > 1). Comprehensive selection model constructed using factor loading matrix: Y = Σ (weight × standardized trait value).

**Visualization**: Correlation heatmaps and PCA biplots generated using GraphPad Prism 9.0 (Graphpad Software, Boston, MA, USA).

## 5. Conclusions

This study establishes a robust, PCA-driven framework for multidimensional trait selection in alfalfa (*Medicago sativa* L.) breeding, effectively addressing key limitations of conventional single-trait approaches. By quantifying six yield-related agronomic traits across parental lines and F_1_/F_2_ hybrid generations, we identified three principal components that together explained 71.14% of the total phenotypic variance. The resulting composite selection model (*Y* = 0.210Z*X*_1_ + 0.360Z*X*_2_ − 0.257Z*X*_3_ − 0.057Z*X*_4_ + 0.194Z*X*_5_ + 0.278Z*X*_6_) enabled the identification of superior F_1_ hybrids, whose F_2_ progenies exhibited significant improvements in dry weight, multifoliolate trait frequency, and yield stability. Notably, the model mitigated generational yield decline and effectively balanced trade-offs among complex traits.

The validated framework not only enhances selection accuracy and breeding efficiency but also demonstrates strong scalability for integration with genomic and environmental datasets. These findings underscore the transformative potential of multivariate selection models in modern forage crop improvement, particularly in response to increasing global demand for livestock feed and the escalating challenges of climate variability.

## Figures and Tables

**Figure 1 plants-14-02906-f001:**
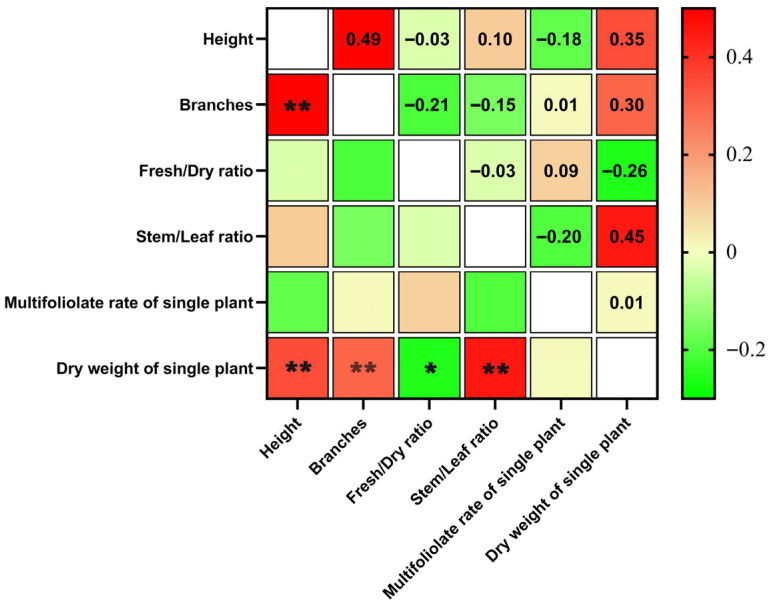
Multi-variate relationships among agronomic traits in alfalfa F_1_ hybrids. Correlation matrix of six agronomic traits in alfalfa F_1_ hybrids derived from reciprocal crosses. Pairwise relationships were assessed using Pearson’s correlation analysis. Asterisks indicate significance levels: ** *p* < 0.01, * *p* < 0.05.

**Figure 2 plants-14-02906-f002:**
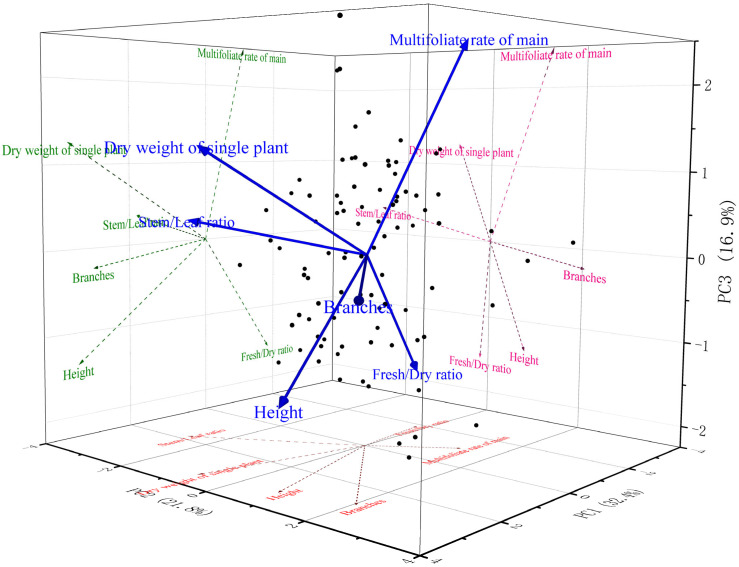
Principal Component Loading Plot of Six Agronomic Traits Based on PCA (PC1–PC3).

**Table 1 plants-14-02906-t001:** Agronomic Performance of Alfalfa Parents and Progeny.

Traits ^1^	Parents ^2^		*p*-Value ^3^	F_1_ Mean
	Paternal Mean (♂)	Maternal Mean (♀)		
Height (cm)	80.85 ± 14.92	72.48 ± 16.23	0.036	75.66 ± 17.64
Branches	7.31 ± 1.90	9.16 ± 3.28	0.008	8.34 ± 3.08
FHR	4.33 ± 0.24	4.04 ± 0.29	0.001	4.16 ± 0.29
LSR	1.64 ± 0.35	1.50 ± 0.29	0.134	1.57 ± 0.31
MF of individual plants (%)	0.00	68.87 ± 4.29	0.000	29.03 ± 5.29
Dry weight of individual plants (g)	119.69 ± 29.17	142.26 ± 43.59	0.034	124.34 ± 47.46

^1^ FHR: fresh/hay yield ratio. LSR: leaf/stem ratio. MF: Multifoliolate trait frequency. ^2^ Results are presented as mean ± standard deviation (SD). ^3^ Statistical significance was determined using two-tailed independent-sample *t*-tests with a significance level of α = 0.05.

**Table 2 plants-14-02906-t002:** Principal Component Analysis Summary: Variance Explained and Extraction Sums of Squared Loadings (PC1–PC3).

Component ^2^	Extraction Sums of Squared Loadings ^1^		
	Total	Percentage of Variance (%)	Cumulative %
1	1.946	32.435	32.435
2	1.306	21.771	54.206
3	1.016	16.937	71.143

^1^ PCA of six agronomic traits identified three principal components (PC1–PC3) with eigenvalues >1, explaining a total of 71.14% of the variance (PC1: 32.43%; PC2: 21.77%; PC3: 16.94%). ^2^ Total (Total Variance Explained): The eigenvalue of each factor, indicating the total amount of variance explained by that factor. Percentage of Variance: The percentage contribution of each factor to the total variance. Cumulative (Cumulative Percentage): The cumulative percentage of total variance explained by the first several factors.

**Table 3 plants-14-02906-t003:** Component Loading Matrix for Six Agronomic Traits Based on PCA.

	Component		
	1	2	3
Z-score (Height)	0.729	0.247	−0.398
Z-score (Branches)	0.645	0.621	−0.071
Z-score (FHR)	−0.412	−0.057	−0.454
Z-score (LSR)	0.410	−0.805	0.102
Z-score (Multifoliolate trait frequency of individual plants)	−0.245	0.395	0.711
Z-score (Dry weight of individual plants)	0.775	−0.229	0.363

**Table 4 plants-14-02906-t004:** Combined Scores of Agronomic Traits in F_1_ Individuals ^1^.

Overall Ranking	Combined Score	Overall Ranking	Combined Score	Overall Ranking	Combined Score
1	35.40	31	0.65	61	−2.54
2	18.89	32	0.62	62	−2.73
3	15.28	33	0.46	63	−2.84
4	13.97	34	0.44	64	−2.92
5	10.82	35	0.33	65	−2.97
6	10.70	36	0.10	66	−3.06
7	10.46	37	0.08	67	−3.15
8	10.31	38	−0.32	68	−3.67
9	10.05	39	−0.34	69	−4.09
10	8.79	40	−0.41	70	−4.26
11	7.79	41	−0.52	71	−4.49
12	7.49	42	−0.53	72	−4.89
13	7.30	43	−0.63	73	−5.55
14	7.10	44	−0.65	74	−5.99
15	6.77	45	−0.66	75	−6.60
16	6.42	46	−0.72	76	−6.78
17	6.15	47	−0.78	77	−7.06
18	6.06	48	−1.02	78	−7.16
19	5.69	49	−1.05	79	−7.71
20	4.88	50	−1.17	80	−7.97
21	4.75	51	−1.44	81	−8.41
22	4.55	52	−1.70	82	−9.37
23	4.01	53	−1.78	83	−9.46
24	3.93	54	−1.83	84	−9.57
25	3.14	55	−1.99	85	−10.26
26	2.86	56	−2.06	86	−10.45
27	1.82	57	−2.17	87	−13.84
28	1.54	58	−2.18	88	−14.19
29	0.95	59	−2.42	89	−14.67
30	0.83	60	−2.52	90	−15.83

^1^ The combined score represents a composite metric derived from the original variables through PCA. It reduces data dimensionality and captures essential patterns by applying specified weighting coefficients according to Formula (5). This integrated score serves as a single indicator for evaluating the overall characteristics of each sample, enabling comparative ranking, assessment, and prediction of yield performance.

**Table 5 plants-14-02906-t005:** Comparative Results of Agronomic Traits of Screened Plants and Hybrid Progeny of The Integrated Evaluation Model ^1^.

	F_1_ Generation Selected Plants	All F_1_ Generation Hybrid Plants	Selected Single Natural Cross F_2_ Generation Plants	Non-Selected Single Natural Cross F_2_ Generation Plants	All F_2_ Generation Plants
Height/cm	86.45 ± 12.04 ^a^	75.66 ± 17.64 ^c^	80.68 ± 9.77 ^b^	69.27 ± 11.09 ^d^	74.98 ± 11.90 ^c^
Branches	11.11 ± 3.02 ^a^	8.34 ± 3.08 ^b^	8.57 ± 1.64 ^b^	7.92 ± 1.81 ^b^	8.24 ± 1.75 ^b^
Ratio of fresh and hay	3.99 ± 0.26 ^b^	4.16 ± 0.29 ^a^	3.83 ± 0.30 ^c^	4.14 ± 0.35 ^a^	3.97 ± 0.36 ^b^
Ratio of stem and leaf	1.60 ± 0.19	1.57 ± 0.31	1.62 ± 0.27	1.59 ± 0.24	1.61 ± 0.25
Multifoliolate trait frequency of individual plants/%	34.68 ± 7.86 ^c^	29.03 ± 5.29 ^c^	50.74 ± 9.92 ^a^	42.71 ± 6.35 ^b^	46.73 ± 8.13 ^ab^
Dry weight of individual plants/g	161.21 ± 45.34 ^a^	124.34 ± 19.46 ^cd^	143.69 ± 44.67 ^b^	112.80 ± 27.65 ^d^	128.24 ± 40.15 ^c^

^1^ Values are presented as means ± standard errors. Different lowercase letters indicate statistically significant differences at *p* < 0.05 based on multiple comparison tests. Identical letters indicate no significant difference (*p* > 0.05). Statistical significance was determined using Duncan’s test at the 95% confidence level.

## Data Availability

The original contributions presented in this study are included in the article/Appendix A. Further inquiries can be directed to the corresponding author.

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
