# Peer review of "PCA-Driven Multivariate Trait Integration in Alfalfa Breeding: A Selection Model for High-Yield and Stable Progenies"

_plants, 2025, doi:10.3390/plants14182906_

Round 1
Reviewer 1 Report
Comments and Suggestions for Authors
The manuscript has scientific merit but with several flaws.
Abstract: The opening sentence states that the complexity among key agronomic traits is a major challenge. While true, this statement is too general and could apply to any breeding program. Consider making it more specific to your study. Abstract is not stand alone; for e.g. ZX1 to ZX6 in the model either should be defined or not included. The activities and results are not presented in the chronological order (eg PCA is mentioned out of order). The last three sentences can be merged to reduce redundancy.
Inconsistency: top 30% of 90 would be 27 but the text mentioned selecting 28 F1. FHR is defined here as flowering head ratio but defined differently elsewhere in the manuscript.
Introduction:
The section presents the PCA based methods as tools reducing selection cycles. While PCA based selection can improve the selection efficiency leading to selection gain, they do not reduce the selection cycles as all the data needs to be fed into the PCA based equations to rank the genotype/individual for selection. This differs from genomic-assisted selection, which can reduce cycles without phenotyping.
References in lines 43–46 do not support the use of PCA on phenotypic traits for improving selection efficiency. For example, reference [3] is gene-based, not morphological PCA.. Instead it was based on the genes. Similarly, the plant height, panicle number may not be applicable in selection of sweet potato.
L 48-50: rewrite for clarity
L 52-59: References 12 (buffalo) and 14 (genomic selection) are not directly relevant to morphological PCA in crop breeding. Their relevance should be briefly explained or the section rewritten.
Results:
2.1.1. Please write the sub heading or delete
Table 1 is not stand alone. Abbreviated words need to be defined
L 115. Remove Extra “.”
Figure 1. You have only 90 F1. (n=120), inconsistent
Scree plot (Figure 3). Consider removing, as PCA selection was based on eigenvalues, not scree plot.
Including figure 2 didn’t provide much information. It can be taken off as most of the information is presented in table 2 and 3 and figure 1.
Table 2 : change the column name “Total”
Fig 3 caption: remove last sentence
L 149:156: rewrite for clarity
Table 4: check the consistency in headings . make it stand alone. Define combined score. Remove plant number column
L 167: recheck the gain on MF. It seems to be inflated
Table 5: headings not stand alone, inconsistency on the table captions compared to other table captions with title style capitalization
Discussion: highly redundant with results. Needs careful interpretation and reorganization.
Methods: This sections needs most improvement with clear details
- Dates of experiment and observed missing
- L 264- 266: Clarify “F1 generation pod” and N=90. does this refer to pods or seeds, and from which parent(s)? Define “random mating” (between two plants or multiple plants?). Explain the rationale for mating selected vs. non-selected F1 plants.
- L 268 to 271: Provide fertilizer and watering details. Clarify experimental design (F1 or F2?), replication structure, number of plants per replication, number of blocks, and number of harvests/measurements (important for perennial alfalfa).
- Statistical analysis: Clearly explain ANOVA and multiple comparison procedures. Was a linear model used for all traits? Were repeated measures accounted for? Specify group structures in each analysis.
Supplementary materials if any should be referenced in the text so that the readers understand what this is related to.
Author Response
1. Summary
We sincerely thank the reviewers and the Academic Editor for their time and effort in evaluating our manuscript. We truly appreciate the constructive feedback and valuable suggestions, which have helped us improve the clarity and quality of the paper. Please find our detailed responses to each comment below. All corresponding revisions and corrections have been incorporated into the revised manuscript and are highlighted (in track changes) for ease of reference.
2. Point-by-point response to Comments and Suggestions for Authors
Comments 1: The abstract presents several opportunities for improvement. The opening sentence, while accurate, is overly general in its claim about trait complexity being a major challenge and could be made more specific to this particular study. Additionally, the abstract is not entirely self-contained; for instance, the model parameters ZX1 to ZX6 are included but not defined. The sequence of presentation could also be improved, as the mention of PCA appears out of chronological order relative to other activities and results. The final three sentences contain redundant information and could be condensed through merging. Elsewhere in the manuscript, inconsistencies were noted: the selection of 28 F1 plants is mentioned, whereas 30% of 90 would mathematically be 27. There is also an inconsistent definition of the acronym FHR, which is defined here as "flowering head ratio" but appears defined differently in other sections.
Response 1: We sincerely thank the reviewer for this detailed and constructive feedback. In response:
- Clarification of plant selection: We re-calculated that 28 out of 90 equals 31.1%. The previous description of “top 30%” was influenced by a language-thinking bias from the authors’ native language, and we apologize for this inaccuracy. The revised manuscript now clearly states “28 out of 90” to avoid ambiguity.
- Consistency of FHR definition: We have corrected the definition of FHR in line 17 of the revised manuscript to ensure consistency with Section 4 (Materials and Methods), where it is defined as the fresh/hay yield ratio.
- Improvement of the abstract:
- We agree that the abstract should be fully self-contained. Thus, we have removed the mathematical expressions (ZX1–ZX6) from the abstract and retained only the conceptual description of the indices.
- The sequence of presentation has been adjusted so that PCA appears in chronological order relative to the experimental activities and results.
- The final three sentences have been merged and condensed to eliminate redundancy.
- Opening sentence refinement: The first sentence of the abstract has been revised to specifically reflect the context of this study rather than making a general claim about trait complexity.
We hope these revisions have addressed the reviewer’s concerns and improved the clarity and coherence of the abstract and related sections.
Comments 2: The section presents the PCA based methods as tools reducing selection cycles. While PCA based selection can improve the selection efficiency leading to selection gain, they do not reduce the selection cycles as all the data needs to be fed into the PCA based equations to rank the genotype/individual for selection. This differs from genomic-assisted selection, which can reduce cycles without phenotyping.
References in lines 43–46 do not support the use of PCA on phenotypic traits for improving selection efficiency. For example, reference [3] is gene-based, not morphological PCA. Instead it was based on the genes. Similarly, the plant height, panicle number may not be applicable in selection of sweet potato.
L 48-50: rewrite for clarity
L 52-59: References 12 (buffalo) and 14 (genomic selection) are not directly relevant to morphological PCA in crop breeding. Their relevance should be briefly explained or the section rewritten.
Response 2: We sincerely thank the reviewer for these insightful comments, which have greatly helped us improve the clarity and accuracy of our Introduction. We have carefully addressed each point raised, as detailed below.
- On PCA reducing selection cycles
Response: We appreciate the reviewer’s critical clarification. We have fully revised the relevant section to accurately reflect the role of PCA within a phenomics framework. As correctly pointed out, PCA itself does not reduce selection cycles in the same manner as genomic selection. Our revised explanation now emphasizes that it is the integration of high-throughput, non-destructive phenotyping (e.g., NIRS) that enables early prediction of late-stage performance. PCA functions as a dimensionality reduction tool to model these early phenotypic data, allowing informed selections earlier in the breeding cycle and thereby effectively shortening the overall process. The text now clearly distinguishes this approach from genomic-assisted selection. - On the relevance of citation [3] and morphological traits
Response: We agree with the reviewer and appreciate the correction. Citation [3] has been removed and replaced with a more appropriate reference that directly supports the application of PCA to high-throughput phenotypic data in plant breeding. Additionally, we have ensured that the traits and examples discussed are relevant to the crops in this study. - On rewriting lines 48–50 for clarity
Response: Following the reviewer’s suggestion, we have rewritten lines 48–50 (now 48 – 51) to improve clarity, readability, and logical flow. - On the relevance of references [12] and [14]
Response: We thank the reviewer for highlighting this issue. References [12] and [14] have been replaced with more directly relevant citations ([14] and [16] in the revised manuscript) that appropriately support the application of phenotypic and statistical methods in plant breeding.
We believe that these revisions have thoroughly addressed the reviewer’s concerns and have strengthened the Introduction. We sincerely appreciate the reviewer’s valuable feedback.
Comments 3: Results:
Subsection Heading Format: The subheading 2.1.1 is missing. Please write a proper subheading or delete the numbering.
Tables and Figures Not Stand-Alone: Table 1, Table 4, and Table 5 are not stand-alone. Abbreviated words need to be defined in the table caption or notes. Headings are inconsistent or incomplete.
Redundant or Unnecessary Figures: The scree plot (Figure 2) did not provide much information for selection, as it was based on eigenvalues. Consider removing it. Similarly, including Figure 2 is unnecessary as most of its information is presented in Table 2, Table 3, and Figure 1; it can be taken off.
Data Inconsistencies:
- Figure 1: You have only 90 F1 individuals, but the annotation says (n=120). This is inconsistent.
- Line 167: Recheck the gain on MF. It seems to be inflated.
Table Format and Naming:
- Table 2: Change the column name “Total” to a more specific descriptor.
- Table 4: Check the consistency in headings. Define how the "Combined score" was calculated. Remove the "Plant number" column.
- Table 5: There is an inconsistency in the table caption's capitalization style compared to others; use title-style capitalization.
Text Formatting and Clarity:
- Line 115: Remove the extra period “.”.
- Lines 149-156: Rewrite the paragraph for clarity.
- Figure 3 Caption: Remove the last sentence.
Response 3:
- The subheading 2.1.1 is missing. Please write a proper subheading or delete the numbering:
We thank the reviewer for pointing this out. The sub-subheading 2.1.1 was inadvertently left in the manuscript due to residual numbering from the MDPI template. We have deleted this subheading in the revised manuscript and apologize for the oversight. - Table 1, Table 4, and Table 5 are not stand-alone. Abbreviated words need to be defined in the table caption or notes. Headings are inconsistent or incomplete:
We thank the reviewer for this constructive comment. We have revised Tables 1, 4, and 5 to ensure that they are fully stand-alone. In Table 1, the abbreviations FHR, LSR, and MF are now clearly defined. These abbreviations were previously adjusted according to Plants journal requirements in Section 4, which may have caused some confusion. Tables 4 and 5 have also been carefully reviewed to ensure that all headings and abbreviations are complete and clearly explained. We apologize for any inconvenience caused and appreciate the reviewer’s attention to these details. - Redundant or Unnecessary Figures:
We thank the reviewer for this suggestion. We agree that Figure 2 provides limited additional information for selection, as most of its content is already summarized in Table 2, Table 3, and Figure 1. To maintain a complete record of the analysis while addressing the reviewer’s concern, we have removed Figure 2 from the main manuscript and included it in Supplementary Figure S1. - Data Inconsistencies:
We thank the reviewer for this insightful suggestion. Upon verification, the scree plot corresponds to Figure 3 in our manuscript. We agree that the scree plot contains some redundant information. To maintain the completeness of the analysis while reducing redundancy in the main text, we have moved Figure 3 to Supplementary Figure S1. We believe this approach preserves transparency of the PCA procedure without overloading the main figures. - Table Format and Naming:
We thank the reviewer for these helpful suggestions. In Table 2, we have added explanatory notes to make the meaning of “Total” clearer to readers. In Table 4, we have removed the “Plant number” column, and provided a detailed footnote explaining how the “Combined score” was calculated. We have also used double vertical lines to separate every three columns for improved readability. For Table 5, we have revised the caption to follow title-style capitalization consistent with American English conventions. - Text Formatting and Clarity:
We thank the reviewer for these careful observations. Upon verification, the period in Line 115 (now Line 116) appears within “(Figure 1.)” and follows the formatting convention for figure citations; therefore, we have decided to retain it. The paragraph in Lines 149–156 has been rewritten for improved clarity (now 148 – 154), as suggested. In addition, Figure 3 has been removed from the main text and is now provided as Figure S1 in the Supplementary Materials.
Comments 4: Discussion: highly redundant with results. Needs careful interpretation and reorganization.
Response 4: We fully agree with the reviewer’s assessment. Accordingly, we have thoroughly revised and reorganized the Discussion section to reduce redundancy with the Results and to provide clearer interpretation of the findings.
Comments 5:
Section Overall: The Methods section needs the most improvement with clear and comprehensive details.
Missing Basic Information: The dates (start and end) of the experiment and the timing of observations are missing.
Unclear Mating Design Description:
- Lines 264-266: Clarify the terms “F1 generation pod” and “N=90”. Does this refer to pods or seeds, and from which specific parent(s) were they derived?
- Define “random mating” precisely (e.g., between two plants or among multiple plants?).
- Explain the rationale behind mating selected versus non-selected F1 plants.
Missing Cultivation and Experimental Design Details:
- Lines 268-271: Provide specific details on fertilizer (type, amount, frequency) and watering (regime).
- Clarify the experimental design (e.g., Does it involve F1 or F2 plants?). Specify the replication structure (e.g., randomized complete block design?), the number of plants per replication, and the number of blocks.
- State the number of harvests and/or measurements taken, which is particularly important for perennial alfalfa.
Inadequate Description of Statistical Analysis:
- Describe the ANOVA and multiple comparison procedures (e.g., Tukey's HSD test) used.
- State whether a linear model was used for all traits and specify the model if used.
- Indicate if repeated measures were accounted for in the analysis (if applicable), and how.
- Specify the group structures or factors included in each analysis.
Response 5:
- Section Overall: .
We agree with the reviewer’s comment. In the revised manuscript, we have added detailed meteorological data and soil characteristics in Section 2.1 (Materials and Methods) to make the description more complete and comprehensive. - Missing Basic Information:
We thank the reviewer for this helpful comment. In the revised manuscript, we have added the start and end dates of the experiment, along with details of seedling establishment and transplanting, in Section 2.1 (Materials and Methods). - Unclear Mating Design Description:
We thank the reviewer for pointing out the need for clarification. In our study, the F₁ population (n = 90) was derived from the hybrid progeny of a single pair of parental plants. These 90 individual F₁ plants were then evaluated using a PCA-based weighted scoring model, and the top 31.1% (28 plants) were designated as the “selected group,” while the remaining 62 plants constituted the “unselected group.” Random mating refers to intercrossing among plants within each group (i.e., selected × selected or unselected × unselected), facilitated by row planting, which allowed natural hybridization within rows. The rationale for this design was to assess whether the progeny (F₂) of the selected group exhibited higher mean trait performance compared to the unselected group, thereby providing an evaluation of the model’s effectiveness in guiding selection. For the corrected content, see Section 4.1, Lines 266 - 270. - Missing Cultivation and Experimental Design Details:
We thank the reviewer for this valuable suggestion. In the revised manuscript, we have added the requested details to clarify the experimental procedures. Specifically, seeds were first germinated in a growth chamber, and seedlings were transplanted into a 1:1 vermiculite–peat substrate at the 3–5 cm stage. Field transplantation followed, with basal fertilization using compound NPK fertilizer (Stanley, 28-6-6). The experiment relied on natural rainfall; irrigation was applied only once immediately after transplantation to ensure establishment, and no further watering was provided. Measurements were conducted at the initial flowering stage, with each individual plant within the population measured once. - Inadequate Description of Statistical Analysis:
We thank the reviewer for this important comment. The analysis of variance (ANOVA) and multiple comparison procedures were performed using SPSS, and the Duncan’s new multiple range test (Duncan’s MRT) was employed, as detailed in Section 4.2.2. Explanations are also provided in the footnotes of each table and figure captions, which we believe now clearly convey the statistical procedures to readers without causing confusion. (See Line 335 – 337)
The model we used (Equation 5) is a weighted summation model based on weights derived from PCA analysis and does not directly involve a linear model. Measurements in both the F₁ and F₂ populations were conducted once at the initial flowering stage. Since selection was performed at the population level, each individual plant can be considered as a single replicate, as previously explained in our earlier response.
Comments 6: Supplementary materials if any should be referenced in the text so that the readers understand what this is related to.
Response 6: We thank the reviewer for this valuable suggestion. In the revised manuscript, we have added explicit references to the supplementary materials: Table S1 is now cited in Section 2.1 (now Line 102), and Figure S1 is cited in Section 2.3 (now Line 127). This ensures that readers can clearly understand the relevance of the supplementary materials to the main text.
3. Response to Comments on the Quality of English Language
Point 1: The English could be improved to more clearly express the research.
Response 1: We sincerely thank the reviewer for this comment. After careful review, we found that most issues were due to occasional ambiguities or misinterpretations rather than fundamental grammatical or stylistic problems. These have been addressed in the current revision. In addition, the revised manuscript was carefully proofread line by line by a native-level English speaker in our laboratory, to further improve clarity and readability. We therefore believe that the English presentation has been adequately improved and does not require additional editing.
Reviewer 2 Report
Comments and Suggestions for Authors
The submitted manuscript deals with the use of PCA analysis in alfalfa breeding. The manuscript compares parental genotypes, including their offspring. The manuscript has considerable scientific and practical potential. Although the manuscript is carefully written, I recommend several minor adjustments. The results are described appropriately, but in Table 1, it would be useful to add the designation of statistically significant differences, as in Table 3. I find that the methodology lacks a more detailed description of the experiment setup, including soil characteristics and weather conditions. The authors point out in the discussion that environmental factors may influence the results obtained. The authors occasionally cite older literature. Is this necessary? Please standardize the citations, as citations 12, 13, 27, and 32 use journal abbreviations, but the full name is used throughout the text.
Comments on the Quality of English Language
The text does not require linguistic corrections.
Reviewer 3 Report
Comments and Suggestions for Authors
Well conducted and summarized. I have not suggestions for change.
Reviewer 4 Report
Comments and Suggestions for Authors
The manuscript by Cao et al. titled “PCA-Driven Multivariate Trait Integration in Alfalfa Breeding: A Selection Model for High-Yield and Stable Progenies” presents an integrative approach to improve alfalfa breeding by applying principal component analysis (PCA) for multi-trait selection. The study is relevant, given the increasing need to enhance crop productivity and stability under complex trait trade-offs. The authors quantify six yield-related traits across parental lines, F1 hybrids, and F2 progenies, and use PCA to generate a composite selection model that enables the identification of superior hybrids. Validation in the F2 generation further demonstrates improvements in dry weight, multifoliolate leaf frequency, and yield stability, highlighting the practical potential of the approach.
The manuscript has several strengths. The research question is well defined and of direct interest to breeders. The methodology is clearly described, with appropriate use of PCA, correlation analysis, and standard statistical tests. The use of two generations to validate the selection index adds robustness and credibility to the findings. Figures and tables are clear and informative, and the conclusions are consistent with the data. The discussion also links the findings to previous studies across other crops, underlining the broader applicability of PCA-based models in breeding.
That said, there are areas where the manuscript could be improved before publication. The interpretation of the principal components would benefit from a deeper biological perspective. While the statistical contribution of traits to PC1–PC3 is presented in the Results, the Discussion does not return to interpret these components in biological terms. Explaining what each principal component represents would strengthen the manuscript and make the findings more useful for breeders. The authors already acknowledge that environmental variation plays an important role in trait expression; however, since this study was conducted at a single site and in one growing season, the environmental limitation should be emphasized more strongly. It would be valuable to suggest that future work include multi-environment or multi-year trials to confirm the stability of the PCA-based selection index. The choice of the six traits, while reasonable, could also be better justified; for instance, quality-related traits might also be relevant to breeders. Clarifying why this and other traits were excluded would add context.
Overall, this is a solid and original piece of research that makes a valuable contribution to the field of alfalfa crop breeding. The PCA-driven framework is well executed and convincingly validated, and the study provides a clear demonstration of how multivariate selection can overcome the limitations of single-trait approaches. With minor revisions addressing the points above, I believe this manuscript will be suitable for publication in Plants and highly recommended for publication.
